# Mitochondrial Oxidative Stress and Mitophagy Activation Contribute to TNF-Dependent Impairment of Myogenesis

**DOI:** 10.3390/antiox12030602

**Published:** 2023-03-01

**Authors:** Daniil A. Chernyavskij, Olga Yu. Pletjushkina, Anastasia V. Kashtanova, Ivan I. Galkin, Anna Karpukhina, Boris V. Chernyak, Yegor S. Vassetzky, Ekaterina N. Popova

**Affiliations:** 1Belozersky Institute of Physico-Chemical Biology, 119992 Moscow, Russia; 2Faculty of Bioengineering and Bioinformatics, Moscow State University, 119992 Moscow, Russia; 3Koltzov Institute of Developmental Biology, 117334 Moscow, Russia; 4CNRS UMR9018, Institut Gustave Roussy, Université Paris-Saclay, 94805 Villejuif, France

**Keywords:** TNF, myogenesis, mitochondria, reactive oxygen species (ROS), mitophagy, antioxidant, SkQ1

## Abstract

Many muscular pathologies are associated with oxidative stress and elevated levels of the tumor necrosis factor (TNF) that cause muscle protein catabolism and impair myogenesis. Myogenesis defects caused by TNF are mediated in part by reactive oxygen species (ROS), including those produced by mitochondria (mitoROS), but the mechanism of their pathological action is not fully understood. We hypothesized that mitoROS act by triggering and enhancing mitophagy, an important tool for remodelling the mitochondrial reticulum during myogenesis. We used three recently developed probes—MitoTracker Orange CM-H2TMRos, mito-QC, and MitoCLox—to study myogenesis in human myoblasts. Induction of myogenesis resulted in a significant increase in mitoROS generation and phospholipid peroxidation in the inner mitochondrial membrane, as well as mitophagy enhancement. Treatment of myoblasts with TNF 24 h before induction of myogenesis resulted in a significant decrease in the myoblast fusion index and myosin heavy chain (MYH2) synthesis. TNF increased the levels of mitoROS, phospholipid peroxidation in the inner mitochondrial membrane and mitophagy at an early stage of differentiation. Trolox and SkQ1 antioxidants partially restored TNF-impaired myogenesis. The general autophagy inducers rapamycin and AICAR, which also stimulate mitophagy, completely blocked myogenesis. The autophagy suppression by the ULK1 inhibitor SBI-0206965 partially restored myogenesis impaired by TNF. Thus, suppression of myogenesis by TNF is associated with a mitoROS-dependent increase in general autophagy and mitophagy.

## 1. Introduction

Impaired muscle regeneration and atrophy of muscle fibers are observed in ageing and many chronic muscle disorders (Duchenne muscular dystrophy (DMD), facioscapulohumeral dystrophy (FSHD), cachexia, etc.) [1,2,3,4]. These disorders are accompanied by oxidative stress and a chronic increase in pro-inflammatory cytokines, including the tumor necrosis factor (TNF) [3,4]. An acute, self-limiting physiological inflammatory response is required for muscle repair after injury, while chronic inflammation impairs repair and causes muscle wasting [4]. Ectopic expression of a secreted form of the murine TNF causes cachexia and impaired muscle repair after injury [5].

High doses of TNF impair myogenic differentiation in cultured myoblasts [6,7,8,9]. TNF also induces muscle protein catabolism in mature myotubes [10,11], as well as apoptosis in myoblasts [7] and myotubes [12]. TNF activates several mechanisms leading to inhibition of myogenic differentiation, some of which are dependent on reactive oxygen species (ROS) [9]. ROS-dependent suppression of myogenesis is partly related to NFkB activation [9,13]. TNF also activates other redox-sensitive mechanisms of myogenesis suppression that are insufficiently studied [9,14].

ROS regulate myogenic differentiation [15,16,17]; however, their role in muscle regeneration and myogenic differentiation is contradictory. ROS levels increase during myoblast differentiation due to a metabolic switch from glycolysis to oxidative phosphorylation and an increase in the number of mitochondria [18]. Moreover, the antioxidant defense of myoblasts and myotubes is significantly less efficient compared to resting muscle stem cells [19]. ROS are necessary for normal myogenic differentiation [20,21,22]. On the other hand, an increased level of ROS is characteristic of many muscle disorders, and an excessive amount of ROS disrupts myogenesis [3,17,23]. Fine tuning of the redox balance is essential for myogenesis. Expression of antioxidant enzymes is increased in response to oxidative stress during myogenic differentiation [21,24]. Knockout of the transcription factor NRF2, which regulates the expression of the oxidative stress response genes, impairs myogenesis and muscle repair [25].

The features of ROS generation in skeletal muscles are described in detail in several reviews [15,19,26]. Briefly, there are several major sources of superoxide radical (O·−2) in skeletal muscles. First, (O·−2) is formed in mitochondria as a result of nonconjugated electron transfer from complexes I and III in the electron transport chain (ETC). Second, nicotinamide adenine dinucleotide phosphate (NADPH) oxidase (NOX) catalyzes the reduction of O2 to (O·−2) using NADH or NADPH as electron donors. Third, xanthine oxidase (XO) generates (O·−2) as a by-product of the oxidation of hypoxanthine to xanthine and uric acid. Fourth, (O·−2) is produced in muscle lipoxygenases (LOX) during the dioxygenation of arachidonic acid, which is released from membrane lipids due to the activity of phospholipase A2. (O·−2) has a relatively long half-life and does not react directly with proteins, carbohydrates or nucleic acids, but can serve as a substrate for the formation of secondary ROS.

Mitochondria are one of the main sources of ROS in muscles [17]. Mitochondrial dysfunction is characteristic of muscle pathologies and is associated with excessive production of ROS that contributes to dysregulation of myogenesis [15,16,17]. Mitochondrial ROS (mitoROS) are capable of activating NFkB in various cell types, including muscle cells [27,28,29]. Application of an efficient mitochondria-targeted antioxidant SkQ1 prevented TNF-induced activation of NFkB and the inflammatory response of endothelial cells [27,28,29,30]. In the cell model of facioscapulohumeral dystrophy (FSHD), SkQ1 significantly reduced excessive oxidative stress and morphological defects in myotube formation [31].

Non-mitochondrial sources of ROS, in particular NADPH oxidases NOX2 and NOX4, also participate in the regulation of myogenic differentiation [20,32,33]. Interestingly, activation of NOX2 in endothelium and in neutrophils depends on mitoROS [34,35,36,37]. Moreover, ROS produced by these enzymes in response to inflammatory mediators cause mitochondrial dysfunction leading to excessive mitoROS production, thus forming a vicious cycle [38].

Oxidative stress, like other stresses such as starvation, hypoxia, ER stress, etc., is a powerful stimulator of canonical macroautophagy. A common target of these stresses is the Unc-51-like kinase 1 (ULK1) complex, which triggers phagophore nucleation and the classic autophagy pathway involving Beclin 1 (BECN1), several autophagy-related genes (ATG), and LC3 [39,40]. Autophagy plays an important role in muscle regeneration and maintenance of homeostasis of this metabolically active tissue [39,40]. Autophagy is increased during satellite cell activation and myogenic differentiation in vivo and in vitro, and its blockage leads to impaired myogenesis [40]. The activation of autophagy in the cellular model of myogenesis is temporary [41]. During myogenesis, selective autophagy of mitochondria (mitophagy) uses the mechanism of classical macroautophagy (reviewed in [42]). AMPK/ULK1 signaling plays an important role in mitophagy induction and muscle regeneration [40]. 

Mitophagy plays an important role in myogenic differentiation and muscle regeneration [43,44]. Presumably, it ensures the removal of old dysfunctional mitochondria from differentiating myoblasts, thus suppressing oxidative stress and apoptosis [41,45]. In addition, mitophagy contributes to the stimulation of mitochondrial biogenesis during myogenic differentiation [41]. On the other hand, excessive stimulation of autophagy/mitophagy by mTOR inhibitors or AMPK activators may impair myogenic differentiation [46,47,48]. Natural protective mechanisms can limit mitophagy due to a decrease in ROS levels. For example, the transcription coactivator PGC-1α which stimulates mitochondrial biogenesis inhibits mitophagy by stimulating the expression of antioxidant enzymes during myogenesis [45]. Antioxidant treatment improves the proliferation of muscle progenitor cells [49] and their capacity to form myotubes and to regenerate damaged muscles [50,51,52,53,54]. Exogenous antioxidants can be considered as possible therapeutic agents for the prevention of myogenesis dysregulation.

Here, we investigated the mechanisms of mito-ROS-dependent disruption of myogenic differentiation under the influence of TNF. We focused on the study of the mitoROS-dependent effect of TNF on myoblasts at the stage of their preparation for fusion. To do this, we added TNF once, 24 h before changing the growth medium to the differentiation medium. Under these conditions, TNF disrupted the myogenesis of immortalized human MB135 myoblasts. TNF further enhanced the generation of mito-ROS and lipid peroxidation of mitochondria and mitophagy on days 0 and 1 of differentiation. The classic antioxidant Trolox and the mitochondria-targeted antioxidant SkQ1 prevented TNF-induced excess mito-ROS, lipid peroxidation, and mitophagy, and partially restored defects in myogenic differentiation. This indicated the key role of mito-ROS in the disruption of myogenesis at its early stage. AMPK/ULK1 signaling plays an important role in the induction of mitophagy during muscle regeneration [55,56]. We have shown that excessive stimulation of AMPK by AICAR leads to disruption of myogenesis, as well as stimulation of autophagy by the mTOR inhibitor rapamycin. At the same time, suppression of AMPK/ULK1 by the SBI 0206965 inhibitor, added once 30 min before TNF, partially restored myogenesis disturbed by this cytokine. This indicated that increased mitophagy may be the cause of TNF-induced impairment of myoblasts’ ability to fuse. Thus, we have shown that TNF inhibits the early stage of differentiation by stimulating mitochondrial ROS production, lipid peroxidation of the mitochondrial inner membrane, and excessive mito-ROS-dependent mitophagy.

## 2. Materials and Methods

### Materials

SkQ1 was synthesized by G.A. Korshunova and N.V. Sumbatyan at the Belozersky Institute of Physico-Chemical Biology. All other reagents, unless otherwise noted, are from Sigma, USA. Culture plastic is produced by Costar, USA Cell cultures. Normal human immortalized myoblasts (MB135) derived from the muscle biopsy of a healthy individual [57] were a kind gift from Dr. Stephen Tapscott. Cells were cultured at 37 °C, 5% CO2 in the growth medium composed of 60% DMEM (PanEco, Moscow, Russia), 25% 199 Medium (PanEco, Russia), and 15% fetal bovine serum (FBS) (HyClone, Logan, USA) supplemented with 0.5 ng/mL bFGF (PanEco, Moscow, Russia), 0.2 ug/mL Dexamethasone (Ellara, Moscow, Russia).

TNF, antioxidants, and inhibitors treatment and myoblast differentiation. Cells were plated onto 12-well cell culture plates (2 × 105 cells per well), or 6-well plates (8 × 105 cells per well), or flux bottom confocal dishes (SPL, USA) (8 × 105 cells per dish) and left overnight for attachment. The next day, the cells were treated with antioxidants (100 μM Trolox, 40 nM SkQ1). After 48 h, human recombinant TNF (50 ng/mL, kind gift of dr. L.N. Shingarova, Institute of Bioorganic Chemistry, Moscow) was added. After another 24 h, cells were washed with DMEM twice, and differentiation was induced by replacing the Growth Medium with a low-serum Differentiation Medium composed of 98% DMEM (PanEco, Russia) and 2% horse serum (Biosera, Cholet, France) supplemented with 10 ug/mL insulin, 5.5 ug/mL transferrin, 6.7 ng/mL sodium selenite (Insulin-Transferrin-Selenium Supplement 50×, PanEco, Russia). Autophagy inhibitor SBI-0206965 (5 μM), or activators rapamycin (100 nM) and AICAR (0.5 μM) were added 30 min or 24 h before TNF (denoted in the figures). Antioxidants were re-added to the differentiation medium.

May–Grunwald Giemsa staining. After 3 days of differentiation, the cells were fixed with 4% PFA for 5 min (Euromedex, Souffelweyersheim, France). The wells with PFA-fixed cells were washed with phosphate-buffered saline (PBS), and stained with 200 μL of May–Grunwald dye for 5 min. Then, 1 mL of PBS was added to the wells without May–Grunwald dye removal and the cells were stained for additional 15 min in the diluted May–Grunwald solution. Afterwards, the wells were washed 3 times with distilled water, stained for 1 h with 1 mL of Giemsa stain (diluted 1:10 in PBS), washed with distilled water again and let dry. All procedures were performed at room temperature. The samples were observed and photographed using the Axio Imager microscope (Zeiss, Oberkochen, Germany). Eight random fields of view were captured for each sample. Fusion Index (FI) was estimated using ImageJ by dividing the number of nuclei inside the myotubes by the total number of nuclei for each sample.

Flow cytometry. For flow cytometry, the cells (non-treated or treated with antioxidants and TNF as described in the “TNF, antioxidants and inhibitors treatment and myoblast differentiation” of Materials and Methods) were incubated with MitoViewGreen probe (100 nM, 30 min), TMRM (200 nM, 15 min), mitoROS indicator MitoTrackerOrange CM-H2TMRos (500 nM, 30 min), or with ratiometric mitochondrial lipid peroxidation indicator MitoCLox (200 nM 4 h) [52,53]. Then, the cells were washed with PBS two times, deattached with Trypsin-EDTA (1/1) solution (PanEco, Russia), centrifuged, and resuspended in 100 uL of PBS (on ice). The flow cytometric measurements were performed on the Amnis^®^ FlowSight^®^ Imaging Flow Cytometer (Luminex, Austin, USA) kindly provided by the Moscow State University Development Program PNR5. The Amnis^®^ FlowSight^®^ Imaging Flow Cytometer was equipped with a 488 nm laser (60 mW) and an SSC laser (10 mW). Data were analyzed in Amnis^®^ IDEAS^®^. Statistical analyses for MitoTrackerOrange CM-H2TMRos and MitoViewGreen stain results were performed for the medians of fluorescence distributions detected in the 560–595 nm or 480–560 nm channel, respectively. To analyze the oxidation of MitoCLox, the ratio of the medians of fluorescence distributions in the green (480–560 nm) and red (595–640 nm) channels was calculated for each sample. The data in the figures are presented as the means of these medians, normalized to the control.

Western blot analysis. Immunoblotting was performed as described previously [26]. The cells were lysed in buffer (62.5 mM Tris-HCl, pH 6.8, 2% SDS, 10% glycerol, 50 mM DTT, 0.01% bromophenol blue). Equal amounts of protein were separated onto 6% SDS polyacrylamide gels (for MYH2) or 12% SDS polyacrylamide gels (and for Glyceraldehyde-3-phosphate dehydrogenase (GAPDH)) and then transferred to PVDF membranes (Amersham, Chicago, IL, USA). Membranes were probed with antibodies against Myosin Heavy Chain (MYH2) (R&D Systems, Minneapolis, MN, USA) and then with HRP conjugated Goat Anti-Mouse IgG (Sigma Aldrich, Burlington, MA, USA). To visualize the peroxidase reaction, the SuperSignal West Dura kit (Thermo Fisher Scientific, Waltham, MA, USA) was used in accordance with the manufacturer’s protocol. Images were obtained using the ChemiDoc™ MP System (Bio-Rad, Hercules, CA, USA). The obtained images were analyzed using ImageLab software (version 5.2.1, Bio-Rad, USA). All data are normalized to the expression level of a housekeeping protein (GAPDH). 

Mitophagy detection. For mitophagy analysis, cells seeded in confocal dishes were transduced with a lentiviral vector containing mitoQC construction [58]. Mito-QC consists of a tandem mCherry-GFP tag located on the outer mitochondrial membrane. The GFP signal (green) is quenched upon delivery of mitochondria to lysosomes, while the mCherry signal (red) is preserved in lysosomes, allowing monitoring and quantification of mitophagy. Transduction was performed the next day after cell plating and 12–15 h before the addition of antioxidants (see “TNF, antioxidants and inhibitors treatment and myoblast differentiation” in “Materials and Methods”). Living cell fluorescent microscopy was performed 3 or more days after transduction. At least 8 random pictures were captured for each sample, measurements were performed for 60 cells on average in each experiment; 3–5 experiments were performed for different points. Mitophagy level was evaluated for fluorescent (effectively transfected) cells by counting the % of cells that contained mitolysosomes. The analysis was performed manually on merged composite images using ImageJ. The images were captured with the same exposition for each channel in each repeat.

Statistical analysis. Statistical analyses were performed in GraphPad Prism 9 using ANOVA test with Dunnet’s correction for multiple comparisons or Mann–Whitney U test. Data are presented as mean ± SD. *p*-values less than 0.05 (*), 0.01 (**), 0.001 (***) and 0.0001 (****) were considered significant. All experiments were performed in no less than 3 biological replicates. The number of biological replicates in each experiment (n) is denoted in corresponding figure legends.

## 3. Results

### 3.1. TNF Suppresses Myogenic Differentiation of MB135 Myoblasts and Stimulates Mitochondrial Oxidative Processes and Mitophagy

Pretreatment of MB135 myoblasts with 50 ng/mL TNF 24 h before induction of myogenesis resulted in a strong decrease in myoblast fusion and myotubes formation (Figure 1A,B), as well as inhibition of myosin heavy chain protein (MYH2), a myotube marker, observed on day 3 of differentiation (Figure 1C–E). At the same time, the addition of TNF led neither to a significant change in the number of myoblasts/myotubes nuclei per field (Figure 1A,E) nor to the appearance of apoptotic cells. Moreover, TNF did not decrease the mitochondrial membrane potential (Figure A1). Thus, the suppression of myogenesis could not be due to the toxic or mitotoxic effect of TNF.

Myogenic differentiation of human MB135 myoblasts on day 1 after induction of differentiation was accompanied by a significant increase in the level of mitoROS, measured by MitoTrackerOrange CM-H2TMRos (Figure 2A,B), and an increase in lipid peroxidation of the mitochondrial inner membrane, measured by a MitoCLox ratiometric fluorescent probe [59,60] (Figure 2C,D). Simultaneously, an increase in mitophagy, measured using a fluorescent mito-QC construct, and a decrease in the number of mitochondria, measured using MitoViewGreen, were detected (Figure 2E–H). All these effects have been previously described in other myoblast cultures and seem to be a part of the mitochondrial reticulum renewal program during myogenic differentiation [15,16,17,41,43,44].

TNF additionally increased the level of mitoROS both before (day 0) and after (day 1) the induction of differentiation (Figure 2A,B). Mitochondrial lipid peroxidation (Figure 2C,D) and the number of cells containing mitolysosomes (ML) were also increased by TNF on day 1 after induction of differentiation. TNF also induced an additional decrease in the number of mitochondria in myoblasts (Figure 2G,H), which indicated an additional stimulation of mitophagy.

### 3.2. Antioxidants Partially Restore Differentiation Impaired by TNF

The antioxidants Trolox (100 mkM, a water-soluble analogue of vitamin E) and SkQ1 (40 nM, the mitochondria-targeted antioxidant, which consists of plastoquinone residue conjugated to the penetrating decyltriphenylphosphonium cation [61]), added first 48 h before TNF and then again in the differentiation medium, significantly reduced both the level of mitochondrial lipid peroxidation (Figure 3A–D) and the level of mitoROS (Figure 3E,F) in both control and TNF-treated cells.

Trolox and SkQ1 added first 48 h before TNF and then again in the differentiation medium partially restored the fusion defects (Figure 4A,B) and MYH2 expression (Figure 4C–E) affected by this cytokine. However, the addition of both antioxidants only at the stage of differentiation was ineffective (Figure A2). In addition, these antioxidants did not restore myogenesis when TNF was added to the differentiation medium (Figure A3).

### 3.3. Antioxidants Restore TNF-Impaired Myogenesis, in Part by Suppressing Excessive Mitophagy

The excess of cells containing ML in the MB135 cell population treated with TNF was decreased by Trolox and SkQ1 antioxidants (Figure 5). The effect of antioxidants was statistically significant on day 1 after the induction, suggesting that restoration of TNF-impaired myogenesis by antioxidants may be at least partially related to suppression of excessive mitophagy.

To test this possibility, we analyzed the effects of autophagy inducers as well as an autophagy inhibitor on myogenic differentiation of MB135 myoblasts. Autophagy inducers rapamycin (mTORC1 inhibitor, 100 nM) and AICAR (AMPK activator, 0,5 μM) added 24 h before differentiation induction caused a decrease in myoblast fusion and MYH2 expression on day 3 of differentiation (Figure 6A–C). On the contrary, the suppression of autophagy and mitophagy by the ULK1 inhibitor (5 μM) added 30 min before TNF led to a partial restoration of TNF-impaired myogenesis and did not affect normal differentiation (Figure 6D–F). These data confirm a possible role of excessive mitophagy in the antimyogenic effect of TNF. It is important to note that the suppression of autophagy during differentiation (when SBI-0206965 was added to differentiation medium (MDM)) led to the suppression of MYH2 expression (Figure 6E,F), which indicated an inhibition of differentiation.

## 4. Discussion

Excessive or chronic inflammation is a common feature of various pathologies characterized by the loss of muscle mass. Proinflammatory cytokines such as TNF initiate intracellular signaling pathways leading to protein breakdown and muscle atrophy. These cytokines can also significantly impair myogenic differentiation essential for skeletal muscle regeneration [6,7,8,9]. The mechanisms of their damaging effects are not fully understood. In vitro experiments allow to separately analyze the effects of cytokines at the stages preceding differentiation and upon differentiation. Here, we investigated the effects of TNF on the myogenesis of human MB 135 myoblasts. TNF inhibited myogenesis both when added at the onset of differentiation and 24 h before the differentiation induction, and when removed, when the growth medium was changed to the differentiation medium. Thus, TNF-treated cells could not complete their myogenic commitment. A similar inhibition of myogenic commitment by TNF in mouse C2C12 myoblasts is associated with prevention of cell cycle exit required for further differentiation [8]. In our model, the addition of TNF to subconfluent MB135 myoblasts monolayer 24 h prior to differentiation induction blocked differentiation (myotube formation and MYH2 expression) without a significant effect on cell proliferation.

The production of mitochondrial ROS accompanies myogenic differentiation and presumably correlates with the metabolic switch from glycolysis to oxidative metabolism [62]. At the same time, mitochondrial dysfunction associated with excessive production of ROS is characteristic of various muscle pathologies [3,17,23]. Using the MitoTracker Orange CM-H2TMRos (a nonfluorescent form of MitoTracker Orange emitting fluorescence when oxidized in the mitochondrial matrix), we demonstrated that myogenic differentiation of MB135 myoblasts was accompanied by an increased production of mitoROS at early stages, which is consistent with the data obtained earlier in myoblast cell lines [18,21,22]. We also analyzed the oxidation of mitochondrial inner membrane phospholipids using MitoCLox, a novel mitochondria-targeted ratiometric fluorescent probe [59,60]. Previously, we observed a fraction of myoblasts with significantly oxidized MitoCLox in MB135 cells [59]. This fraction increased in high-density cultures, indicating that mitochondrial lipid peroxidation was associated with the commitment state of myoblasts. We have now observed a significant increase in mitochondrial lipid peroxidation 24 h after induction of differentiation. TNF slightly but statistically significantly increased both the level of mitoROS and the level of mitochondrial lipid peroxidation after differentiation induction. An additional increase in mitoROS levels induced by TNF was also observed before differentiation induction (day 0).

To study the possible role of mitochondrial ROS production in TNF-dependent myogenesis impairment, we used the mitochondria-targeted antioxidant SkQ1 [61]. SkQ1 effectively scavenged mitoROS, thereby preventing oxidation of MitoTracker Orange CM-H2TMRos and mitochondrial lipid peroxidation. The untargeted antioxidant Trolox also prevented these oxidative events but at much higher concentrations. Both antioxidants prevented the impairment of myogenic differentiation by TNF when added at the proliferation stage 48 h before TNF and then into the differentiation medium. Adding antioxidants only at the stage of differentiation was ineffective. This is consistent with the fact that antioxidants were not able to prevent the effect of TNF added to the differentiation medium. Myoblasts are thus most sensitive to antioxidant treatment at the stages of proliferation/preparation for fusion. We previously observed similar effects of SkQ1 and Trolox in a cell model of FHSD, where oxidative stress and impaired myogenesis were caused by low-level expression of DUX4 in MB135 myoblasts [31]. Both models show that the stage of preparation of myoblasts for fusion (myogenic commitment) is very sensitive to the excessive production of mitoROS.

Increased generation of mito-ROS by dysfunctional mitochondria is characteristic of muscle pathologies [15,16,17]. Excess mito-ROS can stimulate the generation of ROS by cytoplasmic systems. For example, NOX2 activation in neutrophils and endothelium depends on mito-ROS [34,35,36,37]. In turn, cytoplasmic ROS are capable of causing mitochondrial dysfunctions and excessive generation of mito-ROS [38]. The antioxidant Trolox is able to directly remove both mito-ROS and cyto-ROS, while SkQ1 is directly able to remove only mito-ROS. The fact that both antioxidants similarly prevent TNF-induced defects in differentiation points to mito-ROS as the main link in the vicious loop of excessive ROS generation leading to oxidative stress and impaired differentiation.

Antioxidant Tempol, but not SkQ1, reduces the thickness of the formed myotubes in MB135 culture [31]. We have now shown that neither SkQ1 nor Trolox affected myoblast fusion and expression of MYH2. In general, published data on the effect of antioxidants on normal myogenesis are ambiguous, which obviously reflects the complexity and inconsistency of the role of ROS in this process, as well as underestimation of the endogenous antioxidant response of differentiating myoblasts [17,19].

Mitochondrial ROS are able to stimulate selective autophagy of mitochondria (mitophagy), which is often considered as a mechanism to prevent the excessive generation of mitoROS by dysfunctional mitochondria [42,63]. We observed a significant increase in mitophagy 24 h after the induction of MB135 myoblasts differentiation. These observations are consistent with the results of earlier studies on other cultures of differentiating myoblasts [41]. We also observed a stimulation of mitophagy by TNF before (day 0) and, to a lesser extent, after (day 1) induction of differentiation in our model. Stimulation of general autophagy by TNF in mouse C2C12 myoblasts is accompanied by mitochondrial depolarization, ROS generation, and apoptosis [64]. However, in our model, depolarized mitochondria were not observed, probably due to the induction of efficient mitophagy that eliminated dysfunctional mitochondria. Trolox and SkQ1 added 24 h before TNF and then again after induction of differentiation suppressed mitophagy without affecting mitophagy in the absence of TNF, while they reduced mitoROS production and mitochondrial phospholipid peroxidation under the same conditions. Thus, an increased mitoROS production is not essential for mitophagy during normal myogenesis, while TNF-induced excessive mitophagy is largely dependent on mitoROS production in MB135 myoblasts.

Mitochondria removal during muscle injury and after intense exercise is regulated by AMPK/ULK1 signaling [55,56]. This signaling also regulates autophagy in general. Excessive stimulation of AMPK by AICAR led to disruption of myogenesis, as well as stimulation of autophagy by inhibition of mTOR by rapamycin; this is consistent with the previously published data [46,47,48]. Suppression of AMPK/ULK1 with an SBA 0206965 (added to the growth medium 30 min before TNF) partially restored the differentiation defects caused by TNF, without affecting normal myogenesis. These data are consistent with a possible role of excessive mitophagy in TNF-induced disruption of myogenic differentiation. However, it should be taken into account that the substances we used (AICAR, rapamycin, SBA 0206965) also affect the general autophagy, which is involved in the regulation of myogenesis, and also has other side effects. It should also be noted that the addition of SBI 0206965 to the differentiation medium resulted in the blocking of myogenesis, which confirmed the important role of autophagy and mitophagy in myogenesis [40,43,44].

Overall, our results indicate that impairment of myogenesis by TNF is mediated by mitoROS-dependent excessive mitophagy that prevents myoblasts from completing their myogenic commitment/preparation for fusion. The delicate balance between mitophagy induction and prevention of mitophagy overstimulation is important for normal myogenesis.

From a practical point of view, myogenesis is strongly dependent on autophagy; thus, the use of autophagy inhibitors to correct defects in myogenesis associated with excessive mitophagy seems highly questionable. At the same time, the mitochondria-targeted antioxidant SkQ1 does not affect normal mitophagy and myogenesis, but only suppresses excessive mitophagy and thus stimulates myogenesis impaired by the TNF inflammatory cytokine. SkQ1 and similar antioxidants may be potentially useful for the complex therapy of inflammatory muscle pathologies associated with impaired myogenic differentiation.

## Figures and Tables

**Figure 1 antioxidants-12-00602-f001:**
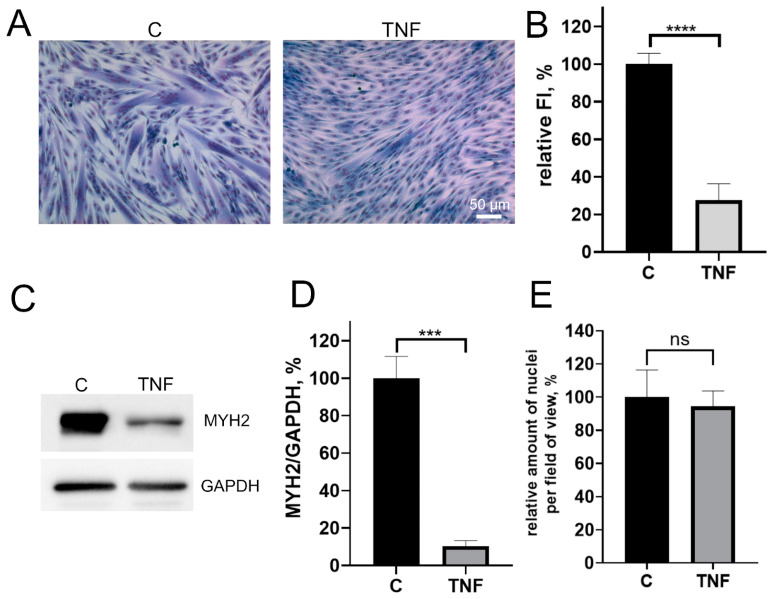
TNF impairs myogenic differentiation of MB135 myoblasts. Cells were treated with 50 ng/mL TNF 24 h before the induction of differentiation. Differentiation was analyzed on day 3 after induction. (**A**) Myotubes were visualized by May–Grunwald staining. Representative images are shown, scale bar: 50 μm. (**B**) Fusion index (FI) was calculated by dividing the number of nuclei inside the myotubes by the total number of nuclei for each sample and FI in the control sample (c) was taken as 100%. (**C**) Representative Western blot showing MYH2 expression. (**D**) Densitometric analysis of Western blots for MYH2 expression in the control and TNF-treated myoblasts. (**E**) Number of myoblast/myotubes nuclei in the control and TNF-treated cultures per field of view. Data are presented as % relative to the untreated control (mean ± SD, ***—*p* < 0.001, ****—*p* < 0.0001, *n* ≥ 3, ns: no significant difference).

**Figure 2 antioxidants-12-00602-f002:**
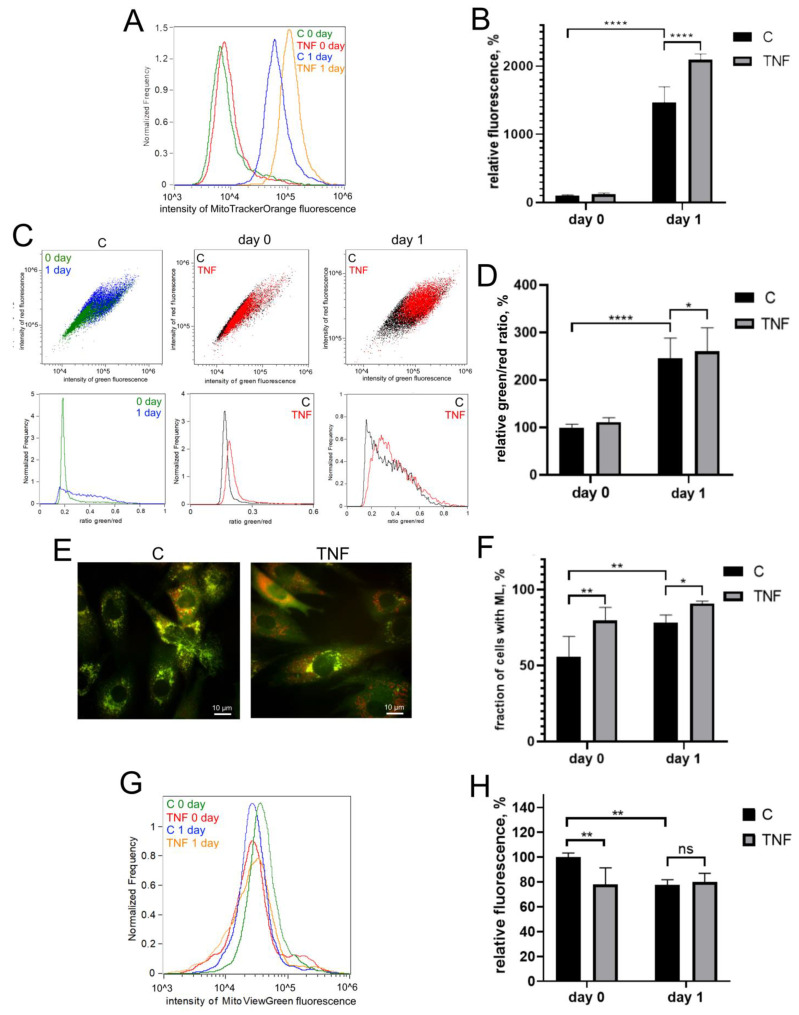
TNF induces an increase in the level of mitoROS, mitochondrial lipid peroxidation and mitophagy in differentiating myoblasts. Cells were treated with 50 ng/mL TNF 24 h before the induction of differentiation. Cells were analyzed before (day 0) and 24 h (day 1) after induction of differentiation. (**A**,**B**) MitoROS levels were measured by flow cytometry with the MitoTracker Orange CM-H2TMRos probe before (day 0) and 24 h (day 1) post differentiation induction in control and TNF-treated myoblasts. (**C**,**D**) Mitochondrial lipid peroxidation was measured by flow cytometry with the MitoCLox ratiometric probe. The ratio of green and red fluorescence was measured on day 0 and day 1 in control and TNF-treated myoblasts. (**E**,**F**) For mitophagy analysis, cells were transduced with the mito-QC construct. Mitolysosomes (ML) are stained in red. Scale bar: 10 μm. The fraction of cells containing ML was measured on day 0 and day 1 in control and TNF-treated myoblasts. (**G**,**H**) Mitochondrial content on day 0 and day 1 in control and TNF-treated myoblasts was measured by flow cytometry with MitoViewGreen. Bars show mean values of median fluorescence intensity ±SD (**B**,**H**) or ratios of median green and red fluorescence intensity ±SD (**D**). Data are presented as % relative to the untreated control at day 0 (mean ± SD, *—*p* < 0.05, **—*p* < 0.01, ****—*p* < 0.0001, *n* ≥ 3, ns: no significant difference).

**Figure 3 antioxidants-12-00602-f003:**
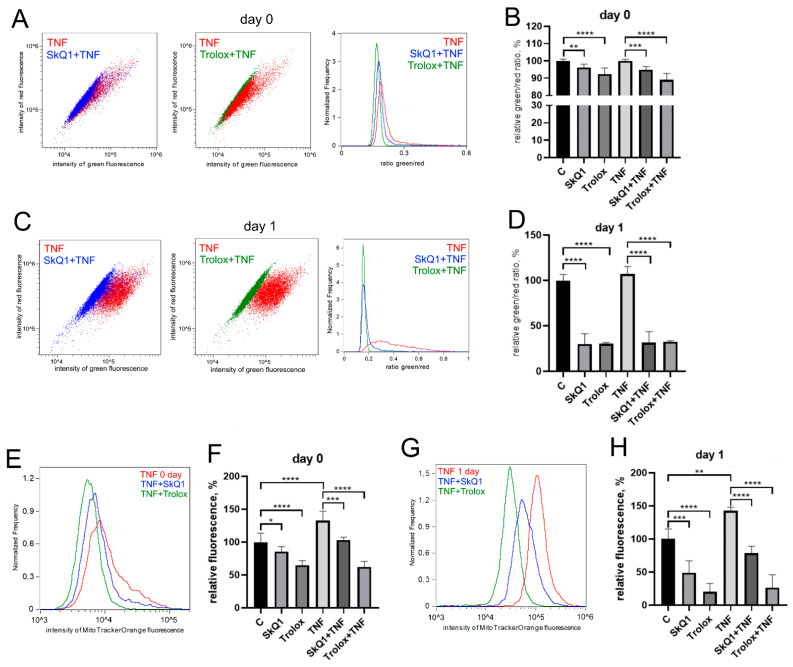
Antioxidants decrease the level of mitoROS and mitochondrial lipid peroxidation in differentiating myoblasts. Cells were treated with 50 ng/mL TNF 24 h before the induction of differentiation. SkQ1 (40 nM) and Trolox (100 µM) antioxidants were added 48 h before TNF and then again into the differentiation medium. Mitochondrial lipid peroxidation was measured by flow cytometry with the MitoCLox ratiometric probe (**A**–**D**); mitoROS levels (**E**,**F**) were measured by flow cytometry with the MitoTracker Orange CM-H2TMRos probe before (Day 0) and 24 h (Day 1) post differentiation induction in control and TNF-treated myoblasts. (**C**,**D**) Mitochondrial lipid peroxidation. (**A**,**C**,**E**,**G**) show typical flow cytometry histograms and scatter plots. (**B**,**D**,**F**,**H**) show mean values of the ratios of median green and red fluorescence intensity ±SD (*—*p* < 0.05, **—*p* < 0.01, ***—*p* < 0.001, ****—*p* < 0.0001, *n* ≥ 3). Data are presented as % relative to untreated control.

**Figure 4 antioxidants-12-00602-f004:**
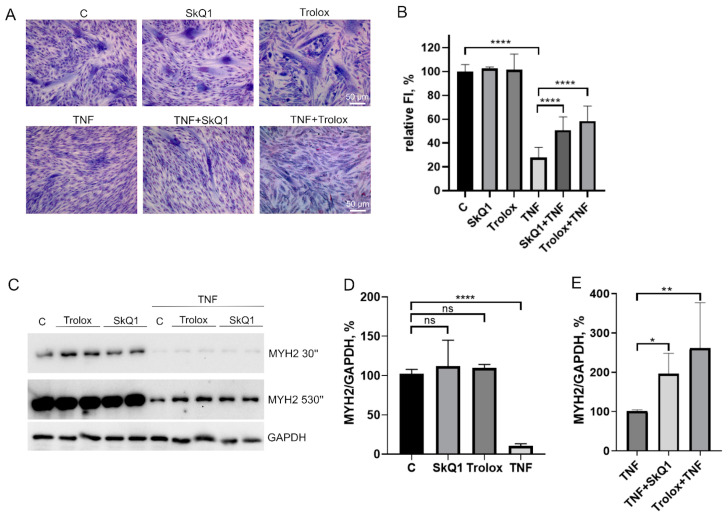
Antioxidants partially restore MB135 differentiation impaired by TNF. Cells were treated with 50 ng/mL TNF 24 h before the induction of differentiation. SkQ1 (40 nM) and Trolox (100µM) antioxidants were added 48 h before TNF and then again into the differentiation medium. Differentiation was analyzed on day 3 after induction. (**A**) Myotubes were visualized by May–Grunwald staining. Representative images are shown, Scale bar: 50 μm. (**B**) Fusion index (FI) was calculated by dividing the number of nuclei inside the myotubes by the total number of nuclei, and FI in the control sample (c) was taken as 100%. (**C**) Western blot analysis of MYH2 expression. (**D**,**E**) Densitometric analysis of Western blots for MYH2 expression. Data are presented as % relative to the untreated control in (**B**,**D**) or as % relative to TNF in (**E**) (mean ± SD, *—*p* < 0.05, **—*p* < 0.01, ****—*p* < 0.0001, *n* ≥ 3, ns: no significant difference).

**Figure 5 antioxidants-12-00602-f005:**
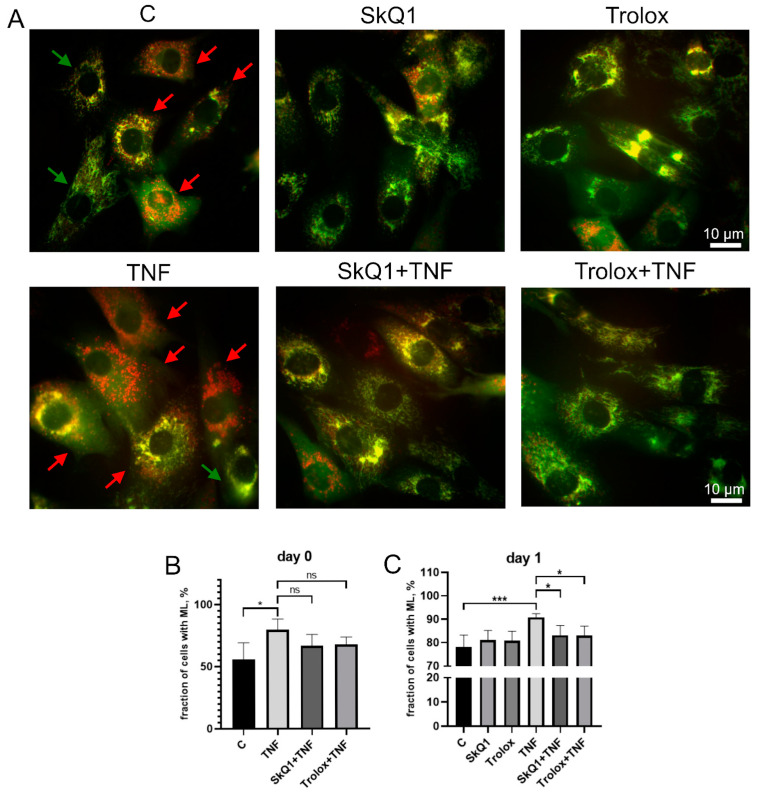
Antioxidants inhibit TNF-induced excessive mitophagy in differentiating myoblasts. Cells were treated with 50 ng/mL TNF 24 h before the induction of differentiation. SkQ1 (40 nM) and Trolox (100 µM) antioxidants were added 48 h before TNF and then again into the differentiation medium. For mitophagy analysis, cells were transduced with the mito-QC construct. Mitolysosomes (ML) are stained in red. Scale bar: 10 μm. The fraction of cells containing ML was measured on the day 0 and day 1 in the control and TNF-treated myoblasts. (**A**) Representative images of cells transduced with the mito-QC construct and treated with antioxidants and TNF, 24 h after differentiation induction. Red arrows indicate cells with ML and green arrows indicate cells without ML. Scale bar: 10 μm. (**B**,**C**) Morphometric analysis of the microscopy images. Data are presented as mean ± SD, *—*p* < 0.05, ***—*p* < 0.001, *n* ≥ 3, ns: no significant difference.

**Figure 6 antioxidants-12-00602-f006:**
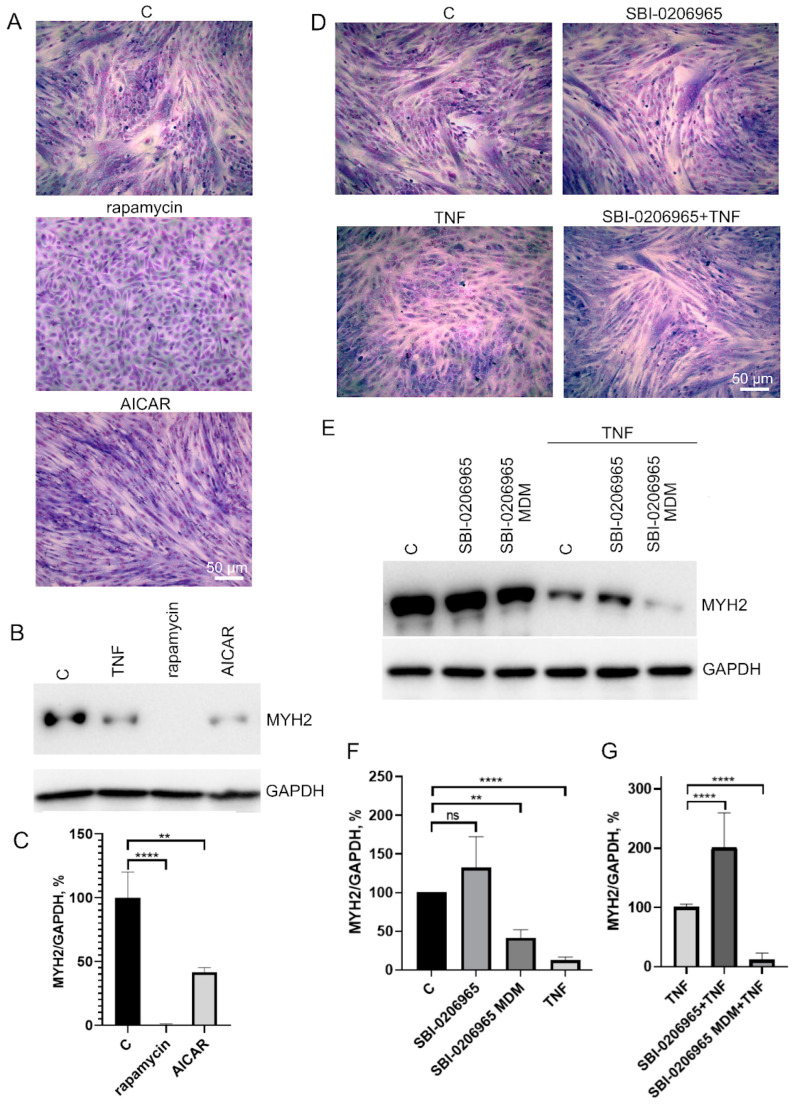
Effects of autophagy inducers (rapamycin and AICAR) and autophagy inhibitor SBI-0206965 on myoblast differentiation. Cells were treated with 50 ng/mL TNF 24 h before the induction of differentiation. Rapamycin (100 nM) and AICAR (0.5 μM) were added 24 h before differentiation induction, SBI-0206965 (5 μM) was added 30 min before TNF or to the differentiation medium (MDM) where indicated. Differentiation was analysed on day 3 after induction. (**A**,**D**) Formation of myotubes as revealed by May–Grunwald staining. Representative images, Scale bar: 50 μm. (**B**,**C**) The effect of autophagy inducers on MYH2 expression. Representative Western blot (**B**) and the results of densitometric analysis (**C**). (**E**–**G**) The effect of the autophagy inhibitor on MYH2 expression in the control and TNF-treated myoblasts. SBI-0206965 was added either 30 min before TNF or in the differentiation medium (MDM). Representative Western blot (**E**) and the results of densitometric analysis of Western blots (**F**,**G**). All the data are presented as mean ± SD, **—*p* < 0.01, ****—*p* < 0.0001, *n* ≥ 3, ns: no significant difference. Data are presented as % relative to untreated control in (**C**,**F**) or as % relative to TNF in (**G**).

## Data Availability

All data generated in this study are contained within the article or Appendix A.

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
