# Peer review of "Mitochondrial Oxidative Stress and Mitophagy Activation Contribute to TNF-Dependent Impairment of Myogenesis"

_antioxidants, 2023, doi:10.3390/antiox12030602_

Round 1

Reviewer 1 Report

This manuscript describes the role of TNF alpha in the development of myogenesis and other mitochondrial functions. While the manuscript is written logically, the data and the presentation can be improved. Below are some of my comments for this paper.

1) The original blots have terrible running of the protein bands on the gel system used by the authors and most of them are very slanted. For eg, Fig 4C in the original blots are not the best. These blots will have to be re-done to ensure high publication standards.

2) The authors discuss induction of autophagy and its effects in Fig 5., however, the manuscript itself has no Fig 5. Can you please include Fig 5 for review. Similarly there is no Fig. A3.

3) The results directed towards the figures in the manuscript has been written in a very confusing manner. The results are not written in order of the figure numbers and therefore is very confusing to go between figures to understand the data.

4) Fig 4A. Images show cells transduced with MitoQC during the differentiation process. Where are the images of the cells showing changes in mitophagy with TNF alpha and antioxidant treatment? They need to be included.

5) Fig 4 E and F. What is the statistical difference between control and TNF treated groups? It appears that the statistical analysis has been done only between a certain subset of groups.

6) The clear goal and objective of the paper did not reflect in this current manuscript. The role of TNF alpha and its effects in myogenesis and differentiation is very well documented. If the authors were trying to show TNF effects in the context of immunology, I believe they would have to show these studies using some type of immune cell system. Otherwise, the study presented is similar to what has been reported in literature before and does contribute extensively to this topic.

Author Response

1) The original blots have terrible running of the protein bands on the gel system used by the authors and most of them are very slanted. For eg, Fig 4C in the original blots are not the best. These blots will have to be re-done to ensure high publication standards.

We used 6% PAAGs  for the determination of MYH (223 kDa). Unfortunately, as the reviewer noted, the images of the 6% stain free gels used for loading control were of poor quality. We simultaneously  run the same samples on 12% gels and used GAPDH (37 kDa) as the load control. These data are now shown in main Figures 1, 2 and 5 and supplementary Figures. We also made appropriate changes to the manuscript. We also included a blot image with a shorter exposure to more accurately estimate the amount of MYH2 in samples without TNF in Figure  2. We have repeated the experiments many times to be sure of our results. Representative blots were selected for publication.

2) The authors discuss induction of autophagy and its effects in Fig 5. However, the manuscript itself has no Fig 5. Can you please include Fig 5 for review. Similarly there is no Fig. A3.

We are extremely sorry for this omission; the Figures 5 and A3 are now added to the MS. 

3) The results directed towards the figures in the manuscript has been written in a very confusing manner. The results are not written in order of the figure numbers and therefore it is very confusing to go between figures to understand the data.

We have changed the sequence of presentation of the results in the figures in accordance with the sequence of their appearance in the text of the Results section. Now, all data concerning the effect of TNF on myoblasts in our model (myoblast fusion, differentiation marker expression, mito-ROS level, mitochondrial lipid peroxidation and mitophagy) are presented separately in Figures 1 and 2. We hope this will make reading the manuscript more comfortable.

4) Fig 4A. Images show cells transduced with MitoQC during the differentiation process. Where are the images of the cells showing changes in mitophagy with TNF alpha and antioxidant treatment? They need to be included.

We have added data on cells treated with TNF and antioxidants to Figure  4A.

5) Fig 4 E and F. What is the statistical difference between control and TNF treated groups? It appears that the statistical analysis has been done only between a certain subset of groups.

We added statistical analysis to Figures 4E and F

6) The clear goal and objective of the paper did not reflect in this current manuscript. The role of TNF alpha and its effects in myogenesis and differentiation is very well documented. If the authors were trying to show TNF effects in the context of immunology, I believe they would have to show these studies using some type of immune cell system. Otherwise, the study presented is similar to what has been reported in literature before and does contribute extensively to this topic.

We have reformulated the aims and results of our work  at the end of the Introduction section as follows: 

Here, we investigated the mechanisms of mito-ROS-dependent disruption of myogenic differentiation under the influence of TNF. We focused on the study of the mitoROS-dependent effect of TNF on myoblasts at the stage of their preparation for fusion. To do this, we added TNF once 24 hours before changing the growth medium to the differentiation medium. Under these conditions, TNF disrupted the myogenesis of immortalised human MB135 myoblasts. TNF further enhanced the generation of mito-ROS and lipid peroxidation of mitochondria and mitophagy on days 0 and 1 of differentiation. The classic antioxidant Trolox and the mitochondria-targeted antioxidant SkQ1 prevented TNF-induced excess mito-ROS, lipid peroxidation, and mitophagy, and partially restored defects in myogenic differentiation. This indicated the key role of mito-ROS in the disruption of myogenesis at its early stage. AMPK/ULK1 signaling plays an important role in the induction of mitophagy during muscle regeneration [61,62]. We have shown that excessive stimulation of AMPK by AICAR leads to disruption of myogenesis, as well as stimulation of autophagy by the mTOR inhibitor rapamycin. At the same time, suppression of AMPK/ULK1 by the SBI 0206965 inhibitor, added once 30 minutes before TNF, partially restored myogenesis disturbed by this cytokine. This indicated that increased mitophagy may be the cause of TNF-induced impairment of myoblasts' ability to fuse. Thus, we have shown that TNF inhibits the early stage of differentiation by stimulating mitochondrial ROS production, lipid peroxidation of the mitochondrial inner membrane, and excessive mito-ROS-dependent mitophagy.

Reviewer 2 Report

The aim of the study is interesting but, in my opinion main, numerous criticisms occurred and must be answered before acceptance in “Antioxidants”.

Please, carefully consider the following criticisms:

1.       Introduction: Rewrite Introduction with a better description of mitochondrial and non-mitochondrial  ROS production

2.       Introduction: Distinct sentences must be used for autophagy, a non-specific process and mitophagy, that is a mitochondria-specific dismantling process in myogenic process. Remember that a high rate of basal autophagy has been reported in muscle, that is different from induced autophagy (Sebastian et al Dev Cell 54,268-281, 2020).

3.       At line 95 insert “excessive” before “mitophagy”

4.       Materials and Methods: use different activators and inhibitors of autophagy. For example, bafilomycin A1 or urolithin see Webb et al Open Access J Neur Neurosurg 42,2017; SBI 026965 has controversial effects see Knudsen et al Int J Mol Sci 21,2344,2020 and Ahwazi D et al Biochem J 478, 2977-2997, 2021.

5.       At lines 288-289 remove the sentence that is most appropriate in the Discussion.

6.       Discussion: Please separate the discussion on autophagy and mitophagy. In the text, please remove Figure numbers that are indicated in Results.

7.       In my opinion, it is not clear why the use of mitochondria-targeted and mitochondria-untargeted antioxidants obtained similar results in this study. I ask the authors to explain this important point.

8.       Lines 378-384 the sentences must be better rewritten because are not fully understandable. However, the authors did not measure autophagy flux nor common autophagy markers like LC3II/I ratio and p62/SQSTM.

9.       References reported at line 385” [61,62,72] “are lacking, please check better Reference list.

10.   Lines 394-395 the sentence must be rewritten to reinforce  the meaning of this study.

Author Response

1.Introduction: Rewrite Introduction with a better description of mitochondrial and non-mitochondrial  ROS production

We have rewritten the Introduction and added a section on ROS production in muscle cells as follows:  

The features of ROS generation in skeletal muscles are described in detail in several reviews [15,19,26]. Briefly, there are several major sources of superoxide radical (O·−2) in skeletal muscles. First, (O·−2) is formed in mitochondria as a result of nonconjugated electron transfer from complexes I and III in the electron transport chain (ETC). Second, nicotinamide adenine dinucleotide phosphate (NADPH) oxidase (NOX) catalyses the reduction of O2 to (O·−2) using NADH or NADPH as electron donors. Third, xanthine oxidase (XO) generates (O·−2) as a by-product of the oxidation of hypoxanthine to xanthine and uric acid. Fourth, (O·−2) is produced in muscle lipoxygenases (LOX) during the dioxygenation of arachidonic acid, which is released from membrane lipids due to the activity of phospholipase A2. (O·−2) has a relatively long half-life and does not react directly with proteins, carbohydrates or nucleic acids, but can serve as a substrate for the formation of secondary ROS.”

2.Introduction: Distinct sentences must be used for autophagy, a non-specific process and mitophagy, that is a mitochondria-specific dismantling process in myogenic process. Remember that a high rate of basal autophagy has been reported in muscle, that is different from induced autophagy (Sebastian et al Dev Cell 54,268-281, 2020).

We have separated the descriptions of autophagy and mitophagy in the Introduction section as follows:

Oxidative stress, like other stresses such as starvation, hypoxia, ER stress, etc., is a powerful stimulator of canonical macroautophagy. A common target of these stresses is the Unc-51-like kinase 1 (ULK1) complex, which triggers phagophore nucleation and the classic autophagy pathway involving Beclin 1 (BECN1), several autophagy-related genes (ATG), and LC3 [39,40]. Autophagy plays an important role in muscle regeneration and maintenance of homeostasis of this metabolically active tissue [39,40]. Autophagy is increased during satellite cell activation and myogenic differentiation in vivo and in vitro, and its blockage leads to impaired myogenesis [40]. The activation of autophagy in the cellular model of myogenesis is temporary [41]. During myogenesis, selective autophagy of mitochondria (mitophagy), which uses the mechanism of classical macroautophagy (reviewed in [42]). AMPK/ULK1 signalling plays an important role in mitophagy induction and muscle regeneration  [40].  Mitophagy plays an important role in myogenic differentiation and muscle regeneration [43,44]. Presumably, it ensures the removal of old dysfunctional mitochondria from differentiating myoblasts, thus suppressing oxidative stress and apoptosis [41,45]. In addition, mitophagy contributes to the stimulation of mitochondrial biogenesis during myogenic differentiation [41]. On the other hand, excessive stimulation of autophagy/mitophagy by mTOR inhibitors or AMPK activators may impair myogenic differentiation [46–48]. Natural protective mechanisms can limit mitophagy due to a decrease in ROS levels. For example, the transcription coactivator PGC-1α which stimulates mitochondrial biogenesis inhibits mitophagy by stimulating the expression of antioxidant enzymes during myogenesis [45]. Antioxidant treatment improves proliferation of muscle progenitor cells [49] and their capacity to form myotubes and to regenerate damaged muscles [50–54]. Exogenous antioxidants can be considered as possible therapeutic agents for the prevention of myogenesis dysregulation.”

  1. At line 95 insert “excessive” before “mitophagy”

Done

  1. Materials and Methods: use different activators and inhibitors of autophagy. For example, bafilomycin A1 or urolithin see Webb et al Open Access J Neur Neurosurg 42,2017; SBI 026965 has controversial effects see Knudsen et al Int J Mol Sci 21,2344,2020 and Ahwazi D et al Biochem J 478, 2977-2997, 2021.

We found that TNF additionally stimulated mitophagy during myoblast differentiation; this effect was dependent on mito-ROS. We needed to test whether an increase in mitophagy affected myogenesis. Unfortunately, there are no specific inhibitors of mitophagy. At the same time, AMPK/ULK1 signalling regulates autophagy and the removal of mitochondria in muscle cells after injury (Call et al. 2017) and exercise (Laker et al. 2017). Therefore, we used an AMPK/ULK1 inhibitor SBI 026965 to suppress mitophagy in our model. We understand that this inhibitor also blocks general autophagy, which is also important for the myogenic differentiation (Laker et al. 2017; Chen et al. 2022). We also understand that, like other autophagy inhibitors, SBI 026965 has side effects unrelated to autophagy and mitophagy that could theoretically affect myogenic differentiation. We have included a discussion of this issue in the "Discussion" section: 

Mitochondrial removal during muscle injury and after intense exercise is regulated by AMPK/ULK1 signalling [55,56]. This signalling also regulates autophagy in general. Excessive stimulation of AMPK by AICAR led to disruption of myogenesis, as well as stimulation of autophagy by inhibition of mTOR by rapamycin; this is consistent with the previously published data [46–48]. Suppression of AMPK/ULK1 with a SBA 0206965 (added to the growth medium 30 min before TNF) partially restored the differentiation defects caused by TNF without affecting normal myogenesis. These data are consistent with a possible role of excessive mitophagy in TNF-induced disruption of myogenic differentiation. However, it should be taken into account that the substances we used (AICAR, rapamycin, SBA 0206965) also affect the general autophagy, which is involved in the regulation of myogenesis, and also have other side effects. It should also be noted that the addition of SBI 0206965 to the differentiation medium resulted in myogenesis arrest, thus confirming an important role of autophagy and mitophagy in myogenesis [40,43,44].“

  1. At lines 288-289 remove the sentence that is most appropriate in the Discussion.

We have removed the sentence in question.

6.Discussion: Please separate the discussion on autophagy and mitophagy. In the text, please remove Figure numbers that are indicated in Results.

We have separated the discussion of autophagy and mitophagy and removed the Figure mentions from the Discussion.

  1. In my opinion, it is not clear why the use of mitochondria-targeted and mitochondria-untargeted

antioxidants obtained similar results in this study. I ask the authors to explain this important point.

Excessive generation of ROS by dysfunctional mitochondria can lead to increased generation of ROS in the cytoplasm. Conversely, cytoplasmic ROS can stimulate the generation of mito-ROS. Removal of excess mitochondrial ROS with SkQ1 can lead to a decrease in the total level of ROS in the cell and the restoration of myogenesis. Trolox removes mitochondrial ROS in the same way as SkQ1, and also directly removes cytoplasmic ROS. From our point of view, the similarity of action of Trolox and SkQ1 shows the paramount importance of mito-ROS in the disruption of myogenesis. At the same time, Trolox restored myogenic differentiation slightly more efficiently than SkQ1 (Fig. 2 of the updated version of the manuscript), which indicates the involvement of non-mitochondrial ROS in the development of the pathological process. We have included a discussion of this issue in the Discussion section as follows:

Increased generation of mito-ROS by dysfunctional mitochondria is characteristic of muscle pathologies  [15–17]. Excess mito-ROS can stimulate the generation of ROS by cytoplasmic systems. For example, NOX2 activation in neutrophils and endothelium depends on mito-ROS [34–37]. In turn, cytoplasmic ROS are capable of causing mitochondrial dysfunctions and excessive generation of mito-ROS [38]. The antioxidant Trolox is able to directly remove both mito-ROS and cyto-ROS, while SkQ1 is directly able to remove only mito-ROS. The fact that both antioxidants similarly prevent TNF-induced defects in differentiation points to mito-ROS as the main link in the vicious loop of excessive ROS generation leading to oxidative stress and impaired differentiation.”

  1. Lines 378-384 the sentences must be better rewritten because are not fully understandable. However, the authors did not measure autophagy flux nor common autophagy markers like LC3II/I ratio and p62/SQSTM.

We have rewritten this part of the discussion as follows:

 “Mitochondria removal during muscle injury and after intense exercise is regulated by AMPK/ULK1 signalling [55,56]. This signalling also regulates autophagy. Excessive stimulation of AMPK by AICAR led to disruption of myogenesis, as well as stimulation of autophagy by inhibition of mTOR by rapamycin, which was consistent with previously published data [46–48]. Suppression of AMPK/ULK1 with a SBA 0206965 (added to the growth medium 30 min before TNF) partially restored the differentiation defects caused by TNF without affecting normal myogenesis. These data are consistent with a possible role of excessive mitophagy in TNF-induced disruption of myogenic differentiation. However, it should be taken into account that the substances we used (AICAR, rapamycin, SBA 0206965) also affect the general autophagy, which is involved in the regulation of myogenesis, and also have other side effects. It should also be noted that the addition of SBI 0206965 to the differentiation medium resulted in blocking of myogenesis, which confirmed the important role of autophagy and mitophagy in myogenesis [40,43,44].”

  1. References reported at line 385” [61,62,72] “are lacking, please check the Reference list.

We are very sorry for this error. These references were lost upon formatting of the  manuscript. We returned them to the updated version and re-checked the final reference list.

  1. Lines 394-395 the sentence must be rewritten to reinforce  the meaning of this study.

We have rewritten this part of the discussion as follows:

At the same time, the mitochondria-targeted antioxidant SkQ1does not affect normal mitophagy and myogenesis, but only suppresses excessive mitophagy and thus stimulates myogenesis impaired by the TNF inflammatory cytokine.”

Reviewer 3 Report

This interesting paper by Chernyavskij DA et al. addressed that TNF-inhibited myogenesis was partially due to excessive mitophagy triggered by TNF-induced mitoROS, using three recently developed probes - MitoTracker Orange CM-H2TMRos, mito-QC, and MitoCLox.  

Did the authors measure mitoROS, the level of mitochondrial lipid peroxidation and mitophagy in differentiating myoblasts when TNF added at the onset of differentiation? If the similar results were observed in these samples as did the use of TNF 24 hours before induction of myoblast differentiation, did the authors perform experiments of 48 hour-treatment of antioxidants followed by TNF + antioxidants in differentiation medium? This reviewer believed that using TNF in the differentiation medium could better represent the setting of aging and diseases.

Please provide the rationale for the doses of all drugs used in this study, including Trolox, SkQ1, TNF, SBI-0206965, rapamycin, and AICAR.  Particularly for TNF, could the high dose of 50 ng/ml be reached in disease conditions?

Please add "excessive" in the last sentence of introduction to make the point clear at the beginning.

Please convert images into the ones with the according color in Figure 4A.

Author Response

Did the authors measure mitoROS, the level of mitochondrial lipid peroxidation and mitophagy in differentiating myoblasts when TNF added at the onset of differentiation? If the similar results were observed in these samples as did the use of TNF 24 hours before induction of myoblast differentiation, did the authors perform experiments of 48 hour-treatment of antioxidants followed by TNF + antioxidants in differentiation medium? This reviewer believed that using TNF in the differentiation medium could better represent the setting of aging and diseases.

We carried out pilot experiments where TNF was added to the myoblast differentiation medium (MDM), and antioxidants were added to the myoblast growth medium (MGM) 72-24 hours before medium change and the MDM. In contrast to the data published on murine C2C12 myoblasts (Langen et al. 2002), antioxidants did not abolish TNF-induced differentiation defects in the human MB135 myoblasts. In addition, TNF added only to the MDM did not increase the mitochondrial lipid peroxidation. Thus, TNF added only to the MGM disrupts the ability of myoblasts to fuse, and this effect is partly associated with an excessive generation of mito-ROS. TNF effects produced after the induction of differentiation are ROS-independent and not amenable by antioxidant therapy. This shows that antioxidants may prove useful in a combined therapy of diseases associated with impaired myogenic commitment of satellite cells.

Please provide the rationale for the doses of all drugs used in this study, including Trolox, SkQ1, TNF, SBI-0206965, rapamycin, and AICAR.  Particularly for TNF, could the high dose of 50 ng/ml be reached in disease conditions?

The range of working concentrations of antioxidants for the MB135 cells was selected by us earlier (Karpukhina et al. 2021; Lyamzaev et al. 2020), it was 2-40ng/ml for SkQ1 and 100-200µM for Trolox. In this model, the concentrations of SkQ1 40 nM and Trolox 100 µM turned out to be the most effective. We selected the concentrations of SBI-0206965, rapamycin, and AICAR earlier when working on other cell cultures, where we assessed LC3 cleavage and cell viability. On myoblasts, we also made sure that the selected concentrations were not toxic. TNF concentration was chosen based on the published data. In myoblast cultures, as a rule, TNF is used at concentrations of tens of ng/ml (Langen et al. 2001; Torrente et al. 2003; Grzelkowska-Kowalczyk and Wieteska-Sk...; Song et al. 2015; Frost et al. 1997). We also took into account that the activity of cytokine preparations can vary greatly depending on their source. We titrated TNF from 1 to 100 ng/mL and chose a concentration of 50 ng/mL. At this concentration, the cytokine was not toxic and, when added to a subconfluent monolayer of myoblasts, did not stimulate their proliferation. In human and animal blood sera, TNF ranges from tenths to tens of pg/ml. It is difficult to say what caused such a difference in concentrations. It is possible that the local concentration of TNF in damaged muscles is higher than in sera. On the other hand, myoblasts after isolation, immortalization, and cultivation may become less sensitive to TNF.

Please add "excessive" in the last sentence of introduction to make the point clear at the beginning.

We have changed this part of the text in the introduction

Please convert images into the ones with the according color in Figure 4A.

In the revised version of the MS, images of cells with mitolysosomes are shown in Fig. 2 E and 5 A. Now we only show the merged images.

Round 2

Reviewer 1 Report

I am satisfied with the changes made by the authors.

Reviewer 2 Report

This version has been ameliorated and authors properly answered my criticismo.

In my opinion, the manuscript deserves to be published.